# Effect of Shading on Physiological Attributes and Proteomic Analysis of Tea during Low Temperatures

**DOI:** 10.3390/plants13010063

**Published:** 2023-12-24

**Authors:** Shah Zaman, Jiazhi Shen, Shuangshuang Wang, Dapeng Song, Hui Wang, Shibo Ding, Xu Pang, Mengqi Wang, Yu Wang, Zhaotang Ding

**Affiliations:** 1Tea Research Institute, Shandong Academy of Agricultural Sciences, Jinan 250100, China; shahzamantea@163.com (S.Z.); shenjiazhitea@163.com (J.S.); wangshuang0103@163.com (S.W.); 2School of Tea and Coffee & School of Bioinformatics and Engineering, Pu’er University, 6 Xueyuan Road, Pu’er 665000, China; 3International Joint Laboratory of Digital Protection and Germplasm Innovation and Application of China-Laos Tea Tree Resources in Yunnan Province, Pu’er University, 6 Xueyuan Road, Pu’er 665000, China; 4Rizhao Tea Research Institute, Rizhao 276800, China; sdp20073882@163.com (D.S.); wh2009tea@126.com (H.W.); rzcksd@163.com (S.D.); pangxutea2020@126.com (X.P.); lfwangmengqife@126.com (M.W.); 5Tea Research Institute, Qingdao Agricultural University, Qingdao 266109, China; wangyutea@163.com

**Keywords:** shade, tea, physiological attributes, proteomics, low temperature

## Abstract

Shading is an important technique to protect tea plantations under abiotic stresses. In this study, we analyzed the effect of shading (SD60% shade vs. SD0% no-shade) on the physiological attributes and proteomic analysis of tea leaves in November and December during low temperatures. The results revealed that shading protected the tea plants, including their soil plant analysis development (SPAD), photochemical efficiency (Fv/Fm), and nitrogen content (N), in November and December. The proteomics analysis of tea leaves was determined using tandem mass tags (TMT) technology and a total of 7263 proteins were accumulated. Further, statistical analysis and the fold change of significant proteins (FC < 0.67 and FC > 1.5 *p* < 0.05) revealed 14 DAPs, 11 increased and 3 decreased, in November (nCK_vs_nSD60), 20 DAPs, 7 increased and 13 decreased, in December (dCK_vs_dSD60), and 12 DAPs, 3 increased and 9 decreased, in both November and December (nCK_vs_nSD60). These differentially accumulated proteins (DAPs) were dehydrins (DHNs), late-embryogenesis abundant (LEA), thaumatin-like proteins (TLPs), glutathione S-transferase (GSTs), gibberellin-regulated proteins (GAs), proline-rich proteins (PRPs), cold and drought proteins (CORA-like), and early light-induced protein 1, which were found in the cytoplasm, nucleus, chloroplast, extra cell, and plasma membrane, and functioned in catalytic, cellular, stimulus-response, and metabolic pathways. In conclusion, the proliferation of key proteins was triggered by translation and posttranslational modifications, which might sustain membrane permeability in tea cellular compartments and could be responsible for tea protection under shading during low temperatures. This study aimed to investigate the impact of the conventional breeding technique (shading) and modern molecular technologies (proteomics) on tea plants, for the development and protection of new tea cultivars.

## 1. Introduction

Tea (*Camellia sinensis* L.), a woody crop, is one of the world’s most popular beverages. Tea is extensively grown in tropical and subtropical areas and typically cultivated in a rain-fed agricultural system. However, because of its adaptability to the original favorable environment, the tea plant is adversely affected by a variety of abiotic stressors [1]. One of the most important environmental factors that has a negative impact on agricultural areas is low temperature. Because of this, not only is the development of plants being severely hindered, but also the security of food supplies around the world is being placed in jeopardy because of a steep decrease in crop production. The overwintering of tea plants and the growth of spring tea are both negatively impacted by low temperatures, which has become one of the most significant environmental concerns [2].

Low temperatures have a significant impact on the chloroplast, which is the primary organelle affected by cold exposure in plants. The effect of low temperatures on tea cell membranes leads to changes in their structure and shape, the restriction of cell membrane permeability, and damage to the cell membranes in tea leaves [3]. This damage hinders the ability to recuperate from stress due to the loss of cellular integrity. Several studies have been undertaken on the effects of low temperature on tea growth and the development of morphological, physiological, and gene expression characteristics [3,4,5]. Plants emit stress signals and trigger many transcription factors when they perceive low temperatures. These variables contribute to the plant’s capacity to withstand stress and regulate the activation of genes that react to cold and other environmental stimuli [6]. Ultimately, this results in the altering of multiple biological processes, including photosynthesis, signaling, transcription, metabolism, cell wall modification, biochemical changes, physiological adaptation, and metabolic changes [1,3]. Low temperatures pose significant limitations on plants, leading to various changes in their structure, physiology, and biochemistry. Cold acclimation, a process that helps plants adapt to cold conditions, has been extensively studied through genomic and transcript-profiling analyses, providing valuable insights. However, it is now acknowledged that the abundance of mRNA transcripts does not always reflect the actual levels of corresponding proteins. This highlights the need to also consider posttranslational regulation mechanisms in understanding the effects of low temperatures on plants [7]. Low-temperature-responsive genes can influence the regulation of differentially expressed proteins when tea plants are exposed to single or combined stressors [5].

As a result, proteomics provides a complimentary approach between the traditional physiological approach and molecular techniques, particularly for research utilizing non-model plant species when no or few genomic sequencing data are available. Proteomic study demonstrated that many proteins are involved in the plant’s response to low-temperature environments [8]. As per the core dogma of genetics, proteins are essential for carrying out genetic functions and are vital participants in most cellular biological processes. Tandem mass tag (TMT) is a highly effective method for detecting changes in protein abundance caused by abiotic stimuli. The tandem mass tag proteome technique utilizes isotopic labeling and tandem mass spectrometry to facilitate both the qualitative and quantitative investigation of proteins. This approach provides numerous benefits, such as exceptional sensitivity, reliable reproducibility, and the capacity to accurately measure protein levels both comparatively and absolutely. Consequently, it has evolved into a powerful tool for examining the levels of proteins in response to environmental stress [8,9,10].

Proteins, which are vital components of enzymes in the tea plant, gather and store the nutrition and energy of the plants. The control of various proteins in plant metabolic processes mostly pertains to plant metabolism and stress tolerance. Some notable proteins that show differential accumulation are dehydrin, late embryogenesis abundant proteins, glutathione s-transferases, thaumatin, and cold-regulated proteins, which have a significant impact on the growth and development of plants. The response of plants to various abiotic stresses has received considerable interest due to the functional mechanisms of proteins [11]. Plant late embryogenesis abundant proteins are classified into eight groups based on the specific motifs and sequence similarity with dehydrins, and late embryogenesis abundant proteins and their homologous genes have been discovered in various higher plants, such as soybean, rice, grape, and other important horticultural plants [12]. In fact, the transgenic tobacco plants significantly increased cold resistance when overexpressing CuCOR19, a gene that encodes late embryogenesis abundant proteins in citrus [13]. It is also hypothesized that late embryogenesis abundant proteins are required for membrane stabilization and protein aggregation prevention [14]. Under cold stress, a significant induction of most late embryogenesis abundant protein family genes in tea indicated their active involvement in the response of tea plants to cold stress [15].

The effect of shading on tea plants grown during low temperatures can depend on several environmental factors such as the specific tea cultivar, the duration of shade, and the intensity, severity, and duration of the low temperatures [3,16]. In our previous study, we also reported that 60% of shade-sheltered tea leaves during mild temperatures inhibited cold damage and enhanced the cellular attributes during low temperatures. However, not much shade contribution was noticed under extreme cold weather in December during low temperatures [3,16]. Another study revealed that the physiological and comparative proteomic analysis of maize under 60% of shade can differentially impact protein folding, modification, and degradation, which alter the plant physiological, biochemical, and metabolic activities [9]. Multiple investigations have been carried out to examine the physiological, biochemical, and molecular mechanisms involved in plants’ response to low-temperature stress [5,9,11,12]. However, the effect of shade on the physiological parameters and proteome analysis of tea leaves under low-temperature conditions has not yet been investigated.

The aim of this study is to examine the effects of shading on the physiological characteristics and proteomics profiling of tea leaves during low temperatures. The objective is to improve the breeding and innovation program for the growth, protection, and development of new tea cultivars under various abiotic stressors, including low temperatures.

## 2. Results

### 2.1. Effect of Shading on Soil Plant Analysis Development and Determination of Nitrogen Content in November and December during Low Temperatures

The effect of shade on the soil plant analysis development of tea leaves was investigated in this study during low temperatures. In December, during low-temperature conditions, no major differences were detected between the control group and the shading group. However, notable differences were seen between the control group and the shading group in November (nCK_vs_nSD60). Interestingly, it was revealed that 60% shade (Nsd60) led to a slight improvement in SPAD values compared to the control (nCK) plants in November at low temperatures (Figure 1A). The effect of shading on the nitrogen content of tea leaves was also measured in November and December during low temperature. No significant differences were observed between the control group and the shading group in December during low temperatures (dCK_vs_dSD60). However, 60% of shading (Nsd60) increased the content of nitrogen compared to control (nCK) in November during low temperatures (Figure 1C).

### 2.2. Effect of Shading on Photochemical Efficiency in November and December during Low Temperatures

The effect of shading on the photochemical efficiency of tea leaves was also calculated under shade during low-temperature. When compared to the control group, the photochemical efficiency was preserved by 60% shade in November. However, low temperatures affected the photochemical efficiency of the unshaded control plants. The photochemical efficiency was also protected by shading (Dsd60) as compared to the control (dCK) group in December during low temperatures (Figure 1B).

### 2.3. TMT Proteomic Analysis of Tea Leaves

The tender mass tags proteomic study of tea was performed in this project under 60% shade compared to 0% in November and December during low temperatures. The expressed proteins were identified using tandem mass tags labeling methods. A total of 7263 proteins were revealed, with 7254 proteins (Appendix A), having a high profusion, and these 7254 proteins primarily functioned in biological components such as metabolic process, cellular process, and response to stimuli in cellular features. The key proteins were regulated in membrane, cell, cell part, and molecular functions such as catalytic activity and molecular function regulator under both groups during low temperatures (Appendix A, Appendix A). According to the Kyoto Encyclopedia of Genes and Genomes (KEGG) pathway enrichment analysis, the key proteins were mostly expressed in the metabolic pathway (Appendix A, Appendix A).

### 2.4. Effect of Shading on Differentially Accumulated Proteins in November and December during Low Temperatures

The present research evaluated the comparative analysis of differentially accumulated proteins between the two groups. Statistically, the fold change was set to (FC < 0.67 and FC > 1.5 *p* < 0.05) according to [9], considered as significant differentially accumulated proteins in November (nCK_vs_nSD60), December (dCK_vs_dSD60), and both November and December shading groups (nSD60_vs_dSD60), respectively. The numbers of differentially accumulated proteins in the December group are higher than those in the November group. However, under both shades (nSD60_vs_dSD60), the number of differentially accumulated proteins found under nCK_vs_nSD60 were 14 DAPs, with 11 increased and 3 decreased. In dCK_vs_dSD60, the total number of key proteins were 20, including 7 increased and 13 decreased, and under nSD60_vs_dSD60, a total of 9, including 3 increased and 9 decreased, were significantly obtained in both months during low temperatures (Figure 2a, Appendix A). A Venn diagram was created to acquire the overlapped differentially accumulated proteins of all groups (Figure 2b, Appendix A). There were nine proteins which were shared between (nCK_vs_nSD60) and (dCK_vs_dSD60). The homologous protein cluster is annotated, and most differentially accumulated proteins occur in posttranslational modification, chaperones, protein turnover, and general prediction, and some of the protein functions are unknown (Figure 2c, Appendix A). The identified differentially accumulated proteins are allied into nine sub-clusters in all groups during low temperatures (Appendix A, Appendix A).

### 2.5. PCA, GO, and KEGG Enrichment Analysis of Differentially Accumulated Proteins in November and December during Low Temperatures

The principal component analysis (PCA) results showed the satisfactory performance of protein differences between groups of samples and the degree of variability between samples within groups (Figure 3A). Differentially accumulated proteins with gene ontology (GO) annotation were found in shaded and control groups under low temperatures. There were 35 enriched differentially accumulated proteins, 19 of which increased and 16 decreased under (nCK_vs_nSD60). In dCK_vs_dSD60, 47 differentially accumulated proteins were enriched, 32 increased and 14 decreased. The total number of enriched proteins was 29, 14 increased and 15 decreased, under nSD60_vs_dSD60.

Most of the key proteins functioned in metabolic processes, cellular processes, catalytic activity, response to stimulus, and binding (Figure 3B, Appendix A). The Kyoto Encyclopedia of Genes and Genomes analysis, statistically (*p* < 0.05), revealed 16 enriched proteins, including 14 increased and 2 decreased, in 14 enriched pathways under (nCK_vs_nSD60). In dCK_vs_dSD60, the total numbers of enriched proteins were 21, including 10 increased and 11 decreased, in 12 enriched pathways. Under nSD60_vs_dSD60, there were 19 proteins, 8 increased and 11 decreased, in 14 Kyoto Encyclopedia of Genes and Genomes enriched pathways (Figure 3C, Appendix A). These differentially accumulated proteins primarily functioned in metabolic pathways, demonstrating that low temperatures mainly alter metabolism.

### 2.6. Subcellular Localization of Differentially Accumulated Proteins in November and December during Low Temperatures

An understanding of the subcellular localization of differentially accumulated proteins is fundamental. Our findings revealed that 21.43% of proteins were localized in chloroplast, 28.57% were noticed in the cytoplasm, 28.57% were present in the nucleus, and 21.43% were found as extra cells under nCK_vs_nSD60 (Figure 4a). The distribution of protein abundance was as follows: 40% in the chloroplast, 10% in the cytoplasm, 15% in the nucleus, 10% in the plasma membrane, and 25% as extracellular profusion of differentially accumulated proteins (Figure 4b). Within the nSD60_vs_dSD60 dataset, we observe that 41.67% of differentially abundant proteins are localized in the chloroplast, 33.33% are found in the cytoplasm, 8.33% are present in the nucleus, 8.33% are in the plasma membrane, and another 8.33% are intricately associated with the extracellular space (Figure 4c).

Moreover, the numbers of differentially accumulated proteins in subcellular localization were also noted in the present work. A total of 14 differentially accumulated proteins were counted in subcellular localization, including 4 in the cytoplasm, 2 increased and 2 decreased in the nucleus, 2 increased and 1 decreased in chloroplast, while 3 increased and 0 decreased were found as extra cells under nCK_vs_nSD60 (Figure 4d, Appendix A). Under dCK_vs_dSD60, a total of 12 differentially accumulated proteins were counted, including 3 increased and 5 decreased in the chloroplast, 5 decreased in extra cell, 1 increased and 2 decreased in the nucleus, 1 increased and 1 decreased in the cytoplasm, and 2 increased in the plasma membrane (Figure 4e, Appendix A). The total number of differentially accumulated proteins counted in subcellular localization was 12, including 2 increased and 3 decreased in the chloroplast, 1 increased and 3 decreased in cytoplasm, 1 decreased in an extra cell in the nucleus, and 1 found in the plasma membrane (Figure 4f, Appendix A).

### 2.7. Effect of Shading on Differentially Accumulated Proteins in November during Low Temperatures

To understand the regulatory mechanism and functions of essential proteins, we systematically analyzed the differentially accumulated proteins under (nCK_vs_nSD60), (dCK_vs_dSD60), and (nSD60_vs_dSD60) during low temperatures. The regulation expression of key proteins and their homologous genes were dehydrin (HYC85_005495), alcohol dehydrogenase (TEA_026230), late embryogenesis abundant protein (HYC85_021841), glutathione transferase (TEA_023893), protein argonaut (HYC85_015860), thaumatin-like protein, nucleolar protein (HYC85_005506), peptidase_S9_N domain-containing protein (TEA_002616), LOV domain-containing protein (TEA_011435), pectinesterase (HYC85_010975), proline-rich protein (HYC85_019506), gibberellin-regulated protein (HYC85_020672), and cold and drought protein CORA-like (TEA_012760), which were accumulated under (nCK_vs_nSD60) in November during low temperatures. Most of these key proteins were mainly located in the nucleus, cytoplasm, chloroplast, and extra cells, and were enriched in metabolic pathways (Table 1 and Appendix A).

### 2.8. Effect of Shading on Differentially Accumulated Proteins in December during Low Temperatures

Likewise, dehydrin (HYC85_008796), agglutinin domain-containing protein (HYC85_024548), lipoxygenase (HYC85_022566), basic endochitinase-like (TEA_029801), late embryogenesis abundant protein (HYC85_021841), early light-induced protein 1, chloroplast-like (TEA_008781), glutathione transferase (TEA_023893), Protein argonaute 16 (HYC85_015860), thaumatin-like protein, nucleolar protein 6 (HYC85_005506), pentatricopeptide repeat-containing protein (HYC85_024587), gibberellin-regulated protein (HYC85_004657), 1,3-beta-glucan synthase (HYC85_031015), proline-rich protein (HYC85_019506), PGG domain-containing protein (TEA_007391), and cold and drought-protein CORA-like (TEA_012760) were accumulated under (dCK_vs_dSD60) in December during low temperatures. These essential proteins were associated with the nucleus, chloroplast, cytoplasm, extra cell, and plasma membrane, and were enriched in different metabolic pathways (Table 1 and Appendix A).

### 2.9. Effect of Shading on Differentially Accumulated Proteins during Both November and December

We also calculated the differentially accumulated proteins under shade in both November and December (nSD60_vs_dSD60) during low temperatures. Interestingly, we found that most proteins were decreased. The key proteins were ribulose bisphosphate carboxylase large chain (Fragment), dehydrin (HYC85_005495), late embryogenesis abundant protein, glutathione transferase (TEA_023893), amine oxidase (HYC85_012990), thaumatin-like protein, PDZ domain-containing protein (TEA_020688), transmembrane and coiled-coil domain-containing protein 4-like (HYC85_002161), COPII coated ERV domain-containing protein (HYC85_011414), and gibberellin-regulated protein (HYC85_020672), which were found mainly in the cytoplasm, nucleus, chloroplast, extra cell, and in plasma membrane (Table 1 and Appendix A).

### 2.10. qRT-PCR Authentication of Differentially Accumulated Proteins during November and December

We used a qRT-PCR of homologous genes of the differently accumulated proteins enriched in significant metabolic pathways in the control groups and under shade during low temperatures to validate the results of the proteomics data (Figure 5). The effect of shading enhanced the expression of alcohol dehydrogenase (*ADH*); nevertheless, no significant variations were identified across all groups during low temperatures. However, the relative expression level of proteins containing PDZ domains (*PDZ*) exhibited substantial differences between December and November during low-temperature conditions among all groups. The levels of glutathione transferase proteins (*GSTs*) showed no significant differences across all groups in both November and December. However, notable distinctions were observed between the shade and control groups in terms of the levels of cold and drought protein CORA-like (*COR*) during the month of November. The expression of another key early light-induced protein 1, chloroplast-like (*ELIP*), was reduced under shade in December compared to control groups. However, the relative expression levels between control groups and those under shade in November were satisfactory even in low-temperature conditions. The relative expression of another glutathione S-transferase (*GSTs*) was shown to have decreased in both the shading and control groups in November, with no significant changes noticed. However, a significant difference was observed between the shade and control groups in December under low-temperature conditions. The differences in regulations of these six randomly selected proteins and the variations in their relative expression were caused by the transcriptional and translational levels under shade and no-shade (control plants) during low temperatures.

## 3. Discussion

### 3.1. Effect of Shading on Physiological Attributes in November and December during Low Temperatures

The soil plant analysis development is essential for assessing the condition of plant leaves at low temperatures. Here, we observe the notable differences in soil plant analysis development under (nCK_vs_nSD60) in November. However, no significant differences were seen under (dCK_vs_dSD60) (Figure 1A). Photochemical efficiency serves as a dependable indicator of plant adaptability to various stressors [17]. Photochemical efficiency exhibited substantial variations when tea leaves were subjected to shading during low temperatures. Both shade conditions exhibited a significantly higher photochemical efficiency compared to the control. However, in November, photochemical efficiency in control plants was higher than that in control plants in December during low temperature (Figure 1B). This suggests that shading may have a significant effect on strengthening leaves and protecting tissues under low temperatures [16]. Similarly, nitrogen determines the growth and quality of tea leaves during normal and stressful conditions. Photosynthesis and photosynthetic machinery can deliver energy for the specific metabolism of plants, and nitrogen is essential for the synthesis of photosynthetic products, enzymes, and chloroplasts in plants [18]. In our findings, the contents of nitrogen were significantly different under (nCK_vs_nSD60) in November. However, no significant differences were found under (dCK_vs_dSD60) in December during low temperatures (Figure 1C). These findings are consistent with our previous results in which 60% of shade improved the nitrogen contents compared to control plants exposed to low temperatures [3]. Our results are also consistent with recently published data in which the authors reported that the effect of shade improved the tea quality and altered the carbon and nitrogen content for metabolic adjustments [19]. As a result of these changes in the physiological characteristics of tea leaves, such as soil plant analysis development, photochemical efficiency, and nitrogen content under cold temperatures, it can be deduced that when tea leaves are directly exposed to cold weather, the accumulation of ice in the intracellular space causes disruptions in the membrane permeability, and low temperatures also cause damage to the extracellular compartments of tissues. On the other hand, shade may prevent ice from accumulating in the cellular compartments of the tea leaves and maintain the membrane system [3,16].

### 3.2. Effect of Shading on Differentially Accumulated Proteins in November and December during Low Temperatures

Understanding the regulatory mechanism of differentially accumulated proteins under shading and low temperatures can offer valuable insights to improve plant performance. In the present study, the effect of shading on the profusion of dehydrin protein was to increase it under (nCK_vs_nSD60) in November but decrease it under (dCK_vs_dSD60) and (nSD60_vs_dSD60) during low temperature. Interestingly, dehydrin plays an important role in plants subjected to drought, salinity osmotic, and cold stress [20,21]. Dehydrin protects cells against the harm caused by stress-induced dehydration, but the specific mechanism is unknown. It has been proposed that its mode of action involves membrane stabilization by acting as a chaperone [22] by preventing protein aggregation and deactivation during dehydration or under fluctuations in temperature [21,22,23]. In our study, tea leaves were exposed to shade and low temperatures during mild cold under (nCK_vs_nSD60) in November, indicating that shade might protect the membrane fluidity in the tissue compartments of tea in a cooperative manner in cold weather. When tea leaves are exposed to extreme temperatures under (dCK_vs_dSD60) in December, the decreased regulation of dehydrin might not serve to safeguard the compartments of the cell against the harmful effect of freezing weather, which might lead to the disruption of the membrane (Table 1). Dehydrin was found to be decreased under both shade conditions in November and December, but the reasons are unknown. Similar trends were observed in late embryogenesis abundant proteins, which were increased under (nCK_vs_nSD60) in November but decreased under (dCK_vs_dSD60) and (nSD60_vs_dSD60) during low temperatures. We can assume that similar phenomena might also occur during the regulation of late embryogenesis abundant protein. Dehydrin also belongs to the second group of the late embryogenesis abundant protein family [22], and alterations in these essential proteins’ abundance might be due to translation and posttranslational modification during tea plant exposure to both shade and low temperatures. The increased abundance of dehydrin and late embryogenesis proteins conferred a stress tolerance mechanism under various abiotic stressors, such as drought, salinity, osmotic, cold, and freezing temperatures [24], by preventing protein denaturation at a low intracellular water content, and avoiding the production of ice crystals within cells [25].

On the other hand, thaumatin-like proteins were significantly up-regulated under (nCK_vs_nSD60) in November and (dCK_vs_dSD60) in December during low temperatures, and decreased under (nSD60_vs_dSD60) during low temperatures. These intensely sweet-tasting proteins of the West African shrub *Thaumatococcus danielli* documented as thaumatin-like proteins have a small molecular weight (mw) of (20–26 kDa). They are proteins with (Cys) residues that mediate eight intra-molecular disulfide bonds which stabilize the proteins under extreme temperature conditions [26], and the increased abundance of thaumatin-like proteins is well known for biotic and abiotic stresses [27,28]. These key proteins also act as antifreeze proteins that prevent ice formation in intracellular spaces and control ice recrystallization in the apoplast of plant tissues [7]. In fact, glutathione S-transferases were increased under (nCK_vs_nSD60) during low temperatures and decreased under (dCK_vs_dSD60) in December and under both November and December (nSD60_vs_dSD60). Glutathione S-transferases protect cells from oxidative damage by stimulating responsive molecules and conjugating GSH to various hydrophobic and electrophilic substrates in plants [29]. Glutathione S-transferases involve various cellular activities and may protect plants from abiotic stressors [30], including cold stress [31]. The findings indicated that early stress induces regulatory functional redundancy, but the exact role of glutathione S-transferases in stress protection is unknown [32]. In our study, when tea leaves were exposed to (nCK_vs_nSD60) during mild low temperatures, the expression of glutathione S-transferases were increased, which might shield cell organelles by recovering cellular integrity. However, a decrease in glutathione S-transferases protein might be associated with extreme stress during December (nCK_vs_nSD60), indicating that cold stress might distract the cellular organelles and inhibit stress recovery by causing cellular integrity to be lost.

Gibberellin-regulated protein was increased under (nCK_vs_nSD60) in November (Table 1), but decreased under (dCK_vs_dSD60) in December and (nSD60_vs_dSD60) during low temperatures, respectively. Gibberellin is particularly important because it regulates homeostasis, stress responses, and cross-talk between signaling pathways [33]. Our study enriched gibberellin-regulated proteins as glycosylphosphatidylinositol (GPI)-anchor biosynthesis. Less information is available regarding glycosylphosphatidylinositol (GPI) in plants. However, in a recent study, the authors suggested that they discovered galactosyltransferase using UDP-galactose as the donor substrate, which might be responsible for the side chain modification that occurs in the golgi apparatus of plants [34]. Further deep investigations are necessary for a breakthrough of this dogma where gibberellin-regulated proteins regulate glycosylphosphatidylinositol (GPI)-anchor biosynthesis in plants under normal and abiotic stressors. Cold and drought protein was increased under (nCK_vs_nSD60) in November and decreased under (dCK_vs_dSD60) in December. Indeed, no detection is found under (nSD60_vs_dSD60) in both shade conditions. This protein and its encoding gene, *COR* or *CcCDR*, play an essential role in multiple abiotic stresses such as drought, salt, and cold stressors [35,36]. In this study, the profusion of cold and drought protein is enhanced in extra cells, which might be beneficial for the surrounding rigid cell membrane layers to cope with low temperatures. Nevertheless, protein CcCDR-GFP was primarily seen in the nucleus, suggesting that CcCDR reaches the nucleus and interacts with other proteins to regulate its homologous gene expression [36]. Proline-rich protein increased under (nCK_vs_nSD60) in November but decreased under (dCK_vs_dSD60) in December. However, no proline rich-protein was found under (nSD60_vs_dSD60) during low temperatures. This protein carries out extracellular repairing and programming in tea leaves. Based on a recent phylogenetic analysis, the authors reported that genes-encoding peptides of proline rich-proteins had been scattered in the chloroplast, extracellular medium, mitochondria, nucleus, and cytoplasm; nevertheless, the chloroplast and extracellular medium are the main points for their localization [37]. These findings suggest that proline-rich proteins are emitted into the extracellular medium via the signal peptide. Our findings indicate that proline rich-proteins and their homologous gene (HYC85_019506) might be involved in plant immunity and performance, and our results are consistent with the findings of [37,38]. Moving forward, early light-induced protein 1 was only decreased under (dCK_vs_dSD60). This indicates that when the temperature drops and the tea plant is exposed to shading and low temperatures, the chloroplast’s integration of pigments in leaves seems to be damaged. Early light-induced protein 1 gene (ELIP1) probably plays a vital role in incorporating pigments into the fully developed light-harvesting pigment–protein complexes. Similarly, a study on the ultrastructure of leaves revealed that shade was responsible for the destruction of the chloroplast ultrastructure, and additional sunlight intensified the grana and lamellae [9]. Both shade and cold stress in the late stage during low-temperature may deter photosystem PS11, weakening the efficiency of light energy and preventing the profusion of electron transfer-related proteins. Similar phenomena were observed in the measurement of photochemical efficiency under (dCK_vs_dSD60) in December during low temperatures. This may be the result of mild low temperatures at the beginning of winter in November and colder temperatures under (dCK_vs_dSD60) in December during low temperatures (Table 1). Studies show that the increased expression of early light-induced protein 1 was observed in conditions where photo-oxidative stress was exacerbated by the simultaneous effects of intense light and cold temperatures [33,39].

## 4. Materials and Methods

### 4.1. Plant Material and Experimental Setup

To investigate the effect of shade on the physiological attributes and proteomics profiling of tea leaves during low temperatures, we conducted an experiment at the Tea Research Institute (Rizhao, China, 35°514′ N, 119°662′ E) during November and December 2021. The five-year mature tea “Zhong Cha 102” *Camellia sinensis* cultivar was selected for comparative proteomic analysis under shade and no-shade (control plants). To create the shade condition, we used black polythene net curtains as a shade and set up 50 m long rows on a metal fence. The shading intensity was set to 60% using black clothing shade, with a control of 0% shade (Table 2, Figure 1D). The experiment was based on control vs. shade (nCK_vs_nSD60) in November and (dCK_vs_dSD60) in December during low temperatures. The average minimum temperature recorded during the experiment was −11.2 °C, while the average maximum temperature was 19.1 °C (Figure 1E). The first and second harvesting were performed at 11:00 a.m. (24/11) and 11:00 a.m. (23/12), respectively. Thirty samples of top leaves were collected from each group with six replicates for physiological attributes, and the remaining samples were kept in the refrigerator at −80 °C for proteomics analysis.

### 4.2. Soil Plant Analysis Development and Determination of Nitrogen Content

The soil plant analysis development of the tea leaves was measured in this study. In brief, the fully expanded uppermost leaves of randomly selected plants under shading and no-shade were determined using a SPAD 502 Meter (Minolta Corporation, Tokyo, Japan) under low temperatures, and the values of the soil plant analysis development (SPAD) were recorded according to [40]. The nitrogen content of tea leaves under shading and no-shade was also determined according to [3,16].

### 4.3. Photochemical Efficiency Analysis

Photochemical efficiency was also investigated in this work under shading and no-shade during low temperatures by using a portable photosynthesis system (Li-6400XT, LI-COR, Inc., Lincoln, NE, USA). In short, fully expanded leaves from the shoot tip were adapted in the dark for 30 min prior to measurement. The maximum photochemical efficiency of photosystem II (Fv/Fm) was measured according to [3,16].

### 4.4. Protein Extraction and Trypsin Digestion

Protein concentration was determined using a Bio-Rad (Hercules, CA, USA) BCA Protein Assay Kit. Trypsin digestion was performed using the filter-aided sample preparation (FASP) technique developed by Matthias Mann. Each sample’s digest peptides were reconstituted in 40 μL of 0.1% (*v*/*v*) formic acid after being desalted on C18 Cartridges (EmporeTM SPE Cartridges C18 (standard density), bed I.D. 7 mm, volume 3 mL, Sigma (St. Louis, MO, USA)). A total of 200 g of proteins was digested using a filter-assisted sample preparation (FASP Digestion) in 30 μL SDT buffer (4% SDS, 100 mM DTT, 150 mM Tris-HCl pH 8.0). Ultrafiltration was repeated with UA buffer (8 M Urea, 150 mM Tris-HCl pH 8.0) to remove the detergent, DTT, and other small molecules (Microcon units, 10 kD). Iodoacetamide (100 mM IAA in UA buffer) was added to the samples and they were incubated for 30 min at room temperature in the dark to avoid the decrease of cysteine residues. The filters were washed thrice in 100 μL of UA buffer and twice with NH_4_HCO_3_ for at least 25 mM. After desalting on C18 Cartridges (Empore^TM^ SPE Cartridges C18 (standard density), bed I.D. 7 mm, volume 3 mL, Sigma), the protein suspensions were digested overnight at 37 °C with 4 μg trypsin (Promega, Madison, WI, USA) in 40 μL 25 mM NH_4_HCO_3_ buffer. The peptides were then concentrated using centrifugation under vacuum and reconstructed in 40 L of 0. Based on the relative abundance of tryptophan and tyrosine in vertebrate proteins, the extinction coefficient for a 0.1% (g/L) solution was determined to be 1.1, with some modifications.

### 4.5. Fractionation and Identification

A total of 20 μg of protein per sample was mixed with 5X loading water and boiled for 5 min using 12.5% SDS-PAGE gel sorted proteins (constant current 14 mA, 90 min) and Coomassie Blue R-250 stained protein rings. TMT reagent was used to label and reverse-phase fractionate a 100 g peptide mixture from each sample. Thermo Scientific’s High pH Reversed-Phase Peptide Fractionation Kit separated tagged peptides. The dried peptide mixture was reconstituted and acidified with 0.1% TFA solution before loading onto the equilibrated, high-pH reversed-phase fractionation spin column. Low-speed spinning washed the hydrophobic resin column to desalinate the peptides bound to it. The columns then eluted the bound peptides into 10 centrifuged fractions using a step gradient of rising acetonitrile concentrations in a volatile high-pH elution solution. After being desalted on Empore^TM^ SPE Cartridges C18 (standard density bed I.D. 7 mm, volume 3 cc, Sigma), the fractions were concentrated using vacuum centrifugation.

### 4.6. LC-MS/MS Analysis

Simple nLC (Proxeon Biosystems (Roskilde, Denmark), Thermo Fisher Scientific (Waltham, MA, USA) was linked to a Q Exactive mass spectrometer for 60/90/120/240 min for LC-MS/MS analysis. Peptides were dissolved in buffer A (0.1% formic acid), loaded onto a reverse phase trap column (Thermo Scientific Acclaim PepMap100, 100 m*2 cm, Nano Viper C18), and then separated using a linear gradient of buffer B (84% acetonitrile and 0.1% Formic acid). Positive ion mode was selected on the mass spectrometer. The most numerous survey scan precursor ions (300–1800 *m*/*z*) were dynamically chosen for HCD fragmentation using a top 10 data-driven selection procedure. The automatic gain control (AGC) target was set to 3 × 10^6^, and the maximal inject time was set to 10 ms. Dynamic exclusion lasted for 40.0 s. The resolution for survey scans was 70,000 at *m*/*z* 200, the resolution for HCD spectra was 17,500 at *m*/*z* 200, and the isolation width was 2 *m*/*z*. The normalized collision energy was 30 eV, and the under-fill ratio, which specifies the minimum proportion of the target value likely to be attained at maximum fill time, was 0.1%. The instrument was operated with the mode for peptide recognition enabled. Proteomics analysis was performed by Wuhan Metware Biotechnology Co., Ltd. (Wuhan, China).

### 4.7. Analysis of Proteomic Data

The LC-MS/MS study was performed using Thermo Scientific’s Q Exactive mass analyzer in conjunction with Proxeon Biosystems’ Easy nLC, now (Thermo FMS, Waltham, MA, USA). The original data were evaluated using Proteome Discoverer v1.4 and the MASCOT engine (Matrix Science, London, UK; version 2.2). To match spectral data to the Homo sapiens SwissProt database (20,425 entries) and the false database, the TMT n-plex was the starting point for the measurement method. TMT6/10/16 plex on Tyr and oxidation on Met were designated as dynamic modifications with a maximum of five components and a few alterations, whereas carbamidomethyl on Cys, TMT6/10/16 plex on Lys, and the peptide N-terminal were labeled as stabilizing mods with peptide mass set at 20 ppm and particle mass at 0.1 Da. Further identification and quantitative assessment were carried out on proteins that contained at least one unique peptide. The raw data were uploaded to the integrated proteome resources (iProX), with project id (IPX0005636000).

### 4.8. GO and KEGG Enrichment Analysis of Differentially Accumulated Proteins

Gene ontology (GO) and the Kyoto Encyclopedia of Genes and Genomes (KEGG) database were used to annotate the protein pathways of differentially accumulated proteins (DAPs).

### 4.9. Protein Abundance and Gene Expression Analysis Using qRT-PCR

To verify the accuracy of the proteomic data, we selected six identified proteins (DAPs) and analyzed the expression levels of the homologous genes using qRT-PCR. Primers were designed using Beacon Designer 8, and the primer sequences are shown in Appendix A, as described in our previous study [3].

### 4.10. Statistical Analysis

The data were compared using one-way ANOVA, and the least significant difference (LSD) test was employed for multiple comparisons at a significance level of (*p* < 0.05). GraphPad Prism was used to create the figures.

## 5. Conclusions

The present study investigated the effect of shading on the physiological attributes and proteome profile of the leaves of the “Zhong Cha 102” tea cultivar in November and December during low temperatures. The results suggested that shading protected tea leaves by regulating their physiological mechanisms. The TMT-based proteomic analysis of differential accumulated proteins such as dehydrins, late-embryogenesis abundant proteins, thaumatin-like proteins, glutathione s-transferase, gibberellin-regulated proteins, pro-line-rich proteins, cold and drought proteins, and early light-induced protein 1 showed that they were enriched in plant metabolic pathways. These differentially accumulated proteins were regulated by translation and post-translational modifications by maintaining membrane permeability within the cellular compartments of tea leaves. Thus, conventional breeding strategies, such as “shading”, and molecular techniques, like “proteomics”, are essential for enhancing the growth, protection, and development of new tea cultivars under abiotic stresses, including low temperatures.

## Figures and Tables

**Figure 1 plants-13-00063-f001:**
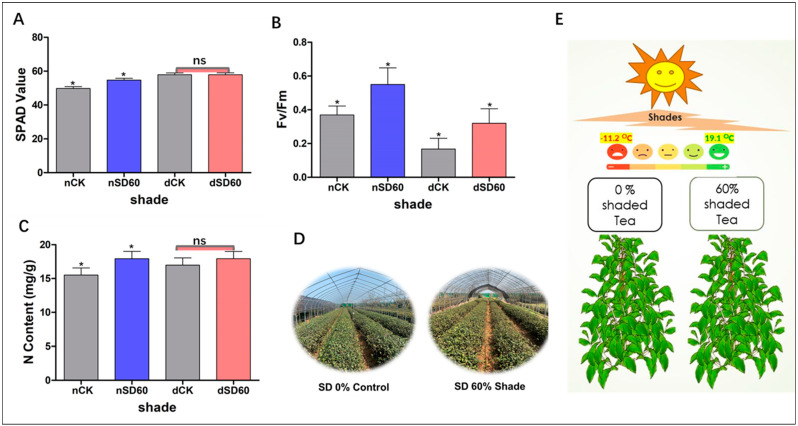
Physiological changes of tea plants under shade and no-shade conditions in November and December during low-temperature. (**A**) Soil plant analysis development. (**B**) Photochemical efficiency. (**C**) Nitrogen content. (**D**) 0% shade vs. SD60% shade intensity. (**E**) Average temperature. The significance difference is indicated by an asterisk and ns represents no-significance differences among all treatments. Data were represented as means and SD from six biological replicates (* *p* < 0.05).

**Figure 2 plants-13-00063-f002:**
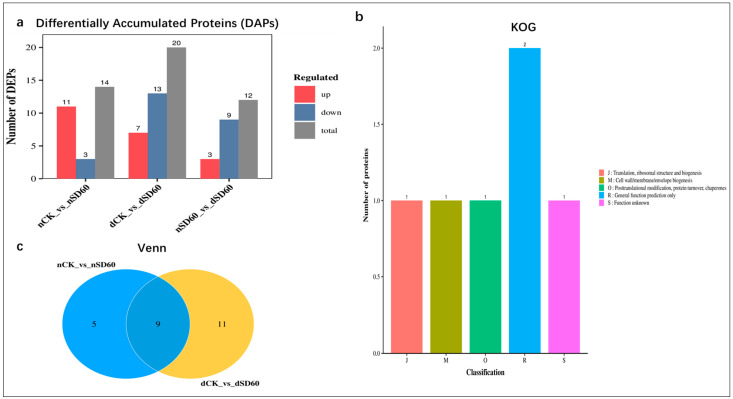
Differentially accumulated proteins (DAPs) in tea plants under (nCK_vs_nSD60) in November, (dCK_vs_dSD60) in December, and (nSD60_vs_dSD60) in both November and December during low temperatures. (**a**) Regulation of proteins. (**b**) Venn diagram analysis of differentially expressed proteins. (**c**) Numbers of proteins in eukaryotic orthologous groups.

**Figure 3 plants-13-00063-f003:**
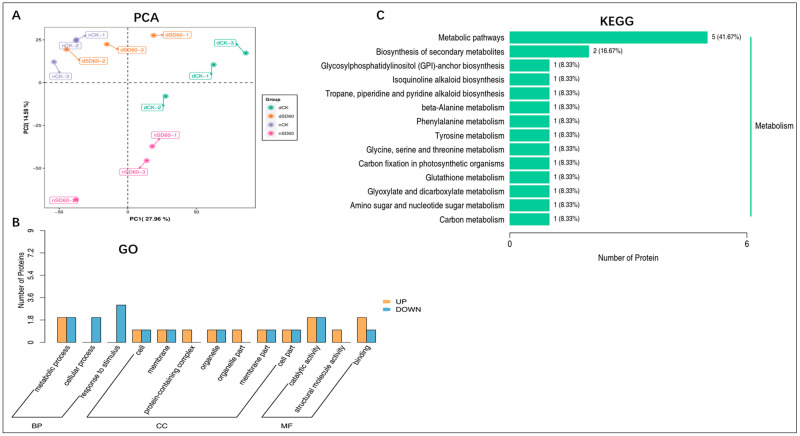
Differentially accumulated proteins in tea plants under (nCK_vs_nSD60) in November, (dCK_vs_dSD60) in December, and (nSD60_vs_dSD60) in both November and December during low temperatures. (**A**) Principal component analysis. (**B**) Gene ontology annotation of differentially expressed proteins. (**C**) Kyoto Encyclopedia of Genes and Genomes annotation of differentially expressed proteins.

**Figure 4 plants-13-00063-f004:**
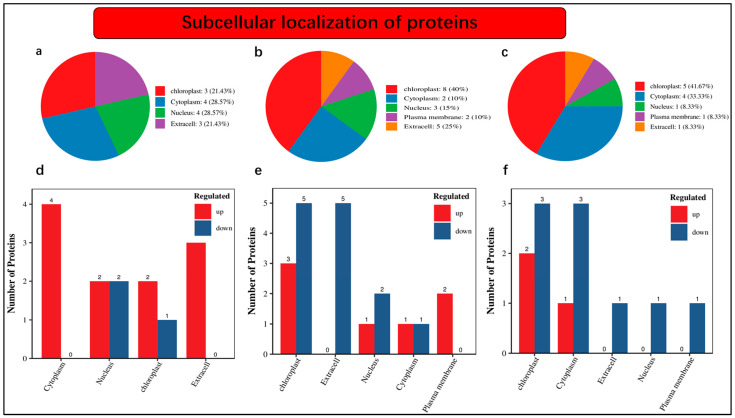
Subcellular localization of differentially accumulated proteins in tea plants under shade and no-shade conditions in November and December during low temperatures. (**a**) Subcellular localization under (nCK_vs_nSD60) in November; (**b**) subcellular localization under (dCK_vs_dSD60) in December; (**c**) subcellular localization under (nSD60_vs_dSD60) in both November and December. (**d**) Numbers of differentially accumulated proteins localized under (nCK_vs_nSD60) in November. (**e**) Numbers of differentially accumulated proteins localized under (dCK_vs_dSD60) in December. (**f**) Numbers of differentially accumulated proteins localized under (nSD60_vs_dSD60) in both November and December.

**Figure 5 plants-13-00063-f005:**
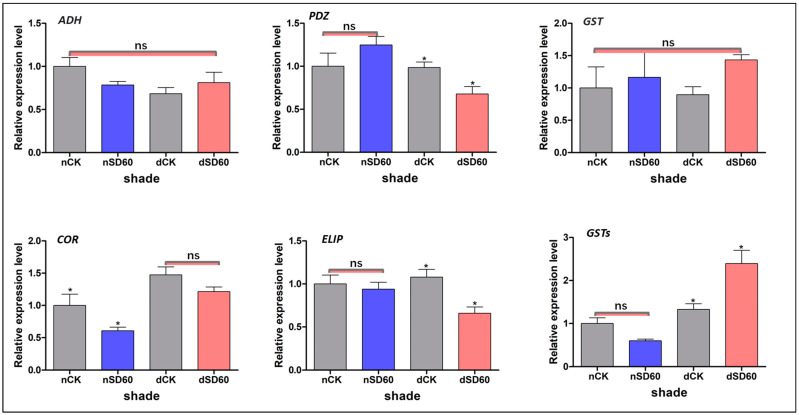
Validation of differentially accumulated proteins such as alcohol dehydrogenase, PDZ domain-containing protein, glutathione transferase protein, cold and drought-regulated protein CORA-like, early light-induced protein 1, chloroplast-like, and the relative expression of their homologous genes were confirmed using qRT-PCR analysis. The significance difference is indicated by an asterisk and ns represents no-significance differences among all treatments. Data were represented as means and SD from six biological replicates (* *p* < 0.05).

**Table 1 plants-13-00063-t001:** Differentially accumulated proteins (DAPs) in tea under (nCK_vs_nSD60), (dCK_vs_dSD60), and (nSD60_vs_dSD60) during low temperatures.

November (nCK_vs_nSD60)				
Accession	Gene	Fold Changes	Regulation	Description
A0A7J7HZN2	HYC85_005495	1.7165	increased	Dehydrin
A0A4S4ENN3	TEA_026230	1.6532	increased	Alcohol dehydrogenase
G0Z053	un-Known	1.5222	increased	Late embryogenesis abundant protein
A0A7J7GMM6	HYC85_021841	2.1332	increased	Late embryogenesis abundant protein D-29-like
A0A4S4DFQ8	TEA_023893	2.1248	increased	Glutathione s transferase
A0A7J7GXX6	HYC85_015860	0.6634	decreased	Protein argonaute 16
A2T1L9	un-known	2.0542	increased	Thaumatin-like protein
A0A7J7I185	HYC85_005506	1.8254	increased	Nucleolar protein 6
A0A4S4E6S2	TEA_002616	0.6191	decreased	Peptidase_S9_N domain-containing protein
A0A4S4ELM5	TEA_011435	1.5961	increased	LOV domain-containing protein
A0A4V3WME9	TEA_012760	1.782	increased	Cold and drought-regulated protein CORA-like
A0A7J7GQS1	HYC85_020672	1.5366	increased	Gibberellin-regulated protein
A0A7J7HLC6	HYC85_010975	0.6654	decreased	Pectinesterase
A0A7J7GQZ8	HYC85_019506	1.6535	increased	Proline-rich protein
**December (dCK_vs_dSD60)**				
A0A7J7HSW9	HYC85_008796	0.6587	decreased	Dehydrin
A0A7J7G8F3	HYC85_024548	1.7761	increased	Agglutinin domain-containing protein
A0A7J7GKR1	HYC85_022566	1.6203	increased	Lipoxygenase
A0A7J7HZN2	HYC85_005495	0.4469	decreased	Dehydrin
A0A4S4CXP2	TEA_029801	0.6232	decreased	Basic endochitinase-like
A0A7J7GMM6	HYC85_021841	0.5414	decreased	Late embryogenesis abundant protein D-29-like
A0A4V3WM41	TEA_012013	1.5687	increased	Agglutinin domain-containing protein
A0A7J7FX29	HYC85_027665	0.6364	increased	Late embryogenesis abundant protein D-29-like
A0A4S4EFX1	TEA_008781	0.5375	decreased	Early light-induced protein 1, chloroplast-like
A0A4S4DFQ8	TEA_023893	0.6037	decreased	Glutathione s transferase
A0A7J7GXX6	HYC85_015860	1.6243	increased	Protein argonaute 16
A2T1L9	un-known	0.4454	decreased	Thaumatin-like protein
A0A7J7I185	HYC85_005506	2.5856	increased	Nucleolar protein 6
A0A7J7GAY3	HYC85_024587	0.5295	decreased	Pentatrico peptide repeat-containing protein
A0A4V3WME9	TEA_012760	0.3446	decreased	Cold and drought-regulated protein CORA-like
A0A7J7HX48	HYC85_004657	0.6633	decreased	Gibberellin-regulated protein
A0A7J7GQS1	HYC85_020672	0.5496	decreased	Gibberellin-regulated protein
A0A7J7GQZ8	HYC85_019506	0.3688	decreased	Proline-rich protein
A0A7J7FPU1	HYC85_031015	1.6614	increased	1,3-beta-glucan synthase
A0A4S4EBQ7	TEA_007391	1.6698	increased	PGG domain-containing protein
**November and December (nSD60_vs_dSD60)**				
A0A2U8T4Y0	un-known	0.3747	decreased	Ribulose bisphosphate carboxylase large chain (Fragment)
A0A7J7HZN2	HYC85_005495	0.4556	decreased	Dehydrin
G0Z053	un-known	0.6488	decreased	Late embryogenesis abundant protein
A0A7J7GMM6	HYC85_021841	0.4582	decreased	Late embryogenesis abundant protein D-29-like
A0A7J7FX29	HYC85_027665	0.6577	decreased	Late embryogenesis abundant protein D-29-like
A0A4S4DFQ8	TEA_023893	0.5286	decreased	Glutathione transferase
A0A7J7HEP6	HYC85_012990	1.5018	increased	Amine oxidase
A2T1L9	un-known	0.5769	decreased	Thaumatin-like protein
A0A4S4D849	TEA_020688	1.6034	increased	PDZ domain-containing protein
A0A7J7I7E8	HYC85_002161	0.6024	decreased	Transmembrane and coiled-coil domain-containing protein
A0A7J7HAZ1	HYC85_011414	1.6267	increased	COPII coated_ERV domain-containing protein
A0A7J7GQS1	HYC85_020672	0.6555	decreased	Gibberellin-regulated protein

**Table 2 plants-13-00063-t002:** Shade specification, measurement, and PAR before sampling in November and December during low temperatures.

			November	December
Treatments	Shade Cloth Specification	Shade Level (%)	PAR (μmol·m^−2^·s^−1^)	PAR (μmol·m^−2^·s^−1^)
SD0	None	0%	701 a	260 a
SD60	Black polyethylene net curtains	60%	263 b	88 c

Specification as per the manufacturer. Treatments: SD0 as control, SD60% as shading. Single-layer black polyethylene net curtains were purchased from Shouguang lvyuan plastic products factory (Weifang, China). PAR: photosynthetically active radiation. Different letters represent significant differences between the treatments indicated using (LSD) test at (*p* < 0.05).

## Data Availability

The Proteomics data can be found on the integrated proteome resources (iProX) under the project id (IPX0005636000).

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
