# Peer review of "Effect of Shading on Physiological Attributes and Proteomic Analysis of Tea during Low Temperatures"

_plants, 2023, doi:10.3390/plants13010063_

Round 1

Reviewer 1 Report

Comments and Suggestions for Authors

Thanks for allowing me to review this manuscript.

The production of tea is obviously a very important topic for the  indigenous producers and the global economy. This is why new proteomic papers can unravel the science behind the optimal conditions for tea production.

Therefore the material presented in this manuscript covers very material.

However, I was slightly disappointed with the presentation of the authors findings which made the science a bit more difficult to understand.

I think that the manuscript could be improved by close attention to the formulation of sentences, paragraphs and whole sections.

I think that the authors should explain their abbreviations as soon as possible so that the potential readership of the manuscript can understand the science as it unfolds.

I started to go through most of the manuscript to correct grammatical and format mistakes but some of these are better to mention briefly since the same types were made frequently.

Major comments

Please spell out the protein names using small letters and not capitals unless these are named after people or places.

Please try not to start sentences with numbers.

Minor comments

Line 19 I eventually worked out what d and n stood for but originally I was confused as to what CK meant. Please let the reader know what these mean early on !

Line 26 DAPs ?

Line 31 should be cold and drought for consistency

Line 33 should be metabolic

Line 38 should be "modern"

Line 43 rephrase for clarity

line 45 reference 2 mentions adaptations to cold but you also mentioned global warming ? is there a reference to this ? is warming a good think for tea plants ?

line 46 must be ?

line 47 defies ? what does this mean in this context ?

line 57 should ref 2 be mentioned here ?

line 57-63 fix grammar, consider chopping sentence in half for ease of reading

line 64 photosynthesis not capitalised

similar capitalisation mistakes made on protein names in lines 75-79

line 90 perhaps you should start  a new paragraph here

line 92 explain TMT abbreviation first time !

could you also rephrase this sentence

line 90 please structure the introducton to tis paragraph a bit clearer

line 103 could you please rewrite this sentence for clarity

same for line 111-114

line 122 abbreviation for CK ?

line 144 should be tandem mass tags

line 146,  number starts sentence ?

line 147 of these ?

line 149 consider rephrasing for clarity

line 156-160 how did the autors determine the appropriate fold change numbers ?  please give reference .

line 163, was less ?

line 163 number starting sentence

line 184 seems like an orphan sentence , is there more context ?

line 204-207 can you rephrase for clarity

line 205-224 much of this information is already descibed in figure 4, could the text of this paragraph be summarised.

line 237-248 protein names should not be capitalised

Table -1 please fix mentions of Un-Known should be unknown !

line 316 what years ?

line 322 did not fully understand, what does this mean in context ?

Comments on the Quality of English Language

English has to be fixed to make manuscript clearer

Author Response

To the commentators, We would like to express our gratitude to each and every reviewer for providing us with such insightful remarks and suggestions regarding our paper. As a result of diligent reading and revision, we were able to improve our manuscript. It is essential to take into consideration the possibility that some remarks come from two or three different reviewers and overlap or contradict one another. Then, we meticulously added everything in a step-by-step manner. First, we addressed the first comments from the reviewer. Next, we handled the second remarks made by the reviewer, and finally, we answered the third comments and suggestions made by the reviewer. If there is something missing, we ask that you kindly accept our apologies. All comments addressed in red fonts in manuscript. We have made every effort to address all of the helpful recommendations, and we have paid a great deal of attention to each and every one of our respected reviewers in order to improve the overall quality of our work. Once again thank you.

Reviewer 2 Report

Comments and Suggestions for Authors

[Minor Comments] Journal: Plants Title: Effect of Shading on Physiological Attributes and Proteomic analysis of tea during low temperature Manuscript Number: plants-2765768 Remarks: This article primarily examines the latest research on the impact of shade on tea leaves during the months of November and December, when temperatures are low and the leaves experience varying levels of stress. The Authors utilizes a conventional breeding approach (Shade) along with advanced technologies like (TMT-based proteomics) analysis. As tea is derived from leaves, this study on the tea plant is intriguing. Hence, the efficacy of tea might vary based on the environments, and shading has diverse effects on the growth, development and performance of tea. This is interesting study due to the unpredictable weather patterns and the impacts of global warming. Exposure to abiotic stress can adversely affect a plant's growth, development, and overall performance. Hence, it is crucial to employ multiple systematic strategies to safeguard our crops from the potentially catastrophic consequences of abiotic stressors, such as low temperature. The analysis of protein expression and its regulation in metabolic pathways is wellestablished, and the assessment of the current literature offered in this study is a good contribution to the field. However: During the review, a few minor errors were noticed. Consequently, I suggest promptly rectifying these flaws to enhance the value of this study article. Overall Comments: Prior to proceeding, it is recommended to thoroughly read the article in its entirety, starting from the abstract and concluding with the last section. When encountering abbreviations enclosed in round brackets, it is advised to retain them only if they appear once for the first time. The authors referred to differentially accumulated proteins (DAPs) throughout the text, however at some point it is indicated as differentially expressed proteins (DEPs). Please rectify all as (DAPs). The title of the manuscript is visually appealing. However, in the title, authors use the term "lowtemperature," while in the content, they occasionally write "low temperature," "low temperatures," etc. Please adhere to the consistent use of the term "low temperature" throughout the manuscript. On line 18, replace the term "minimal" with "less" or a more suitable word. Modify line -26 by applying the italic formatting to the letter "p". Lines 37 and 38 The growth and development of novel tea cultivars can be enhanced by the utilization of sustainable conventional breeding approaches and modern molecular technology. Authors should include the term "shade" or "shading" in combination with conventional breeding, and the term "genomics" in combination with current molecular technology. The introduction is effectively composed in accordance with the subject matter of this investigation. Remove the heading titled "physiological attributes" from line 116. Modify line 140 to display the letter "p" in italic format. Please revise line 158 by changing the letter "p" to italic format and ensure that this modification is applied consistently across the document. The abbreviation "DEP" has been modified to "DAP". 273rd line and homologous genes. delete this. Modify line 290 by replacing "low temperature" with "low-temperature". Table 1. The headings in the table are displayed in various typefaces. Kindly apply bold formatting to them. For example, the months of December (dCK_vs_dSD60) and November & December (nSD60_vs_dSD60) are not in bold. Line 237 Avoid excessive repetition of the abbreviation. As previously stated, only retain the abbreviation for its initial occurrence. At line 368, replace "DEPs" with "DAP" and provide the full form of "DAP". Line-375 should not include duplicate abbreviations. Modify line 555 by italicizing the term "camellia sinensis". line 688- Modify the term "protected culture" to "sustainable conventional breeding". line 714- Remove the term that is revealed in and rephrase the sentence in a suitable manner. Line 715 This section is not mandatory but can be added to the manuscript if the discussion is unusually long or complex. I am unable to comprehend the nature of this sentence. Kindly double check it. Please carefully revise and correct the errors that have been mentioned above.

Comments on the Quality of English Language

Minor editing of English language required

Author Response

(The authors gave the same response as above.)

Reviewer 3 Report

Comments and Suggestions for Authors

The present study was very important in terms of tea plants in future research. I have several comments to improve this manuscript.

1.       There are many grammar and typo errors found in this article.

2.       The objective of this study is missing.

3.       It should elaborate all acronyms at first-time use.

4.       In the abstract, the method needs more clearly present. The abstract is too long and it's difficult for the audience to get the main findings of this study.

5.       Why did you apply the December and November sampling time?

6.       Authors should focus research gap in the introduction.

7.       In Figure 1 E , is there any phenotypic difference found between 0 and 60% shading in tea plants?

8.       There is no significant between dCK and dSD60 in the case of SPAD value but the same treatment Fv/Fm is significantly different, why?

9.       The discussion is also so long.

10.   Methodology should need to be elaborately explained.

11.   Conclusion rewrite precisely to focus on core findings of the study.

Comments on the Quality of English Language

Need to moderate English ediitng.

Author Response

(The authors gave the same response as above.)

Round 2

Reviewer 3 Report

Comments and Suggestions for Authors

The authors did not modify according to the raised queries. The added as well as deleted modifications should be notified.

1.     The objective is missing in the abstract.

2.     The subheadings of the results must be changed such as 2.1.1. Soil Plant Analysis Development, 2.1.2 Photochemical Efficiency and Nitrogen Determination Analysis and so on….

3.     In Figure 4 , the title inside the figure like subcellular localization needs to modify i.e. subcellular localization protein like that.

4.     In figure 5, mentions the genes acronym and italic form. In the caption write in detail.

5.     Discussion must be reduced and not more than 1500 words.

6.     Expression analysis by qRT-PCR, explained elaborately.

Comments on the Quality of English Language

Must need to modify.

Author Response

Dear Editors and Reviewers,

We deeply value your support and consideration throughout the manuscript evaluation process. We highly appreciate the recommendations and comments provided by reputable reviewers as they assisted us in enhancing the quality of our content. We thoroughly revised the entire manuscript, including the abstract, introduction, results, discussion, conclusion, and references. We improved the quality of the English language and carefully addressed each comment from the reviewer, making necessary modifications to the content.   Thank you for understanding.

  1. The objective is missing in the abstract.

Reply: We appreciate your suggestion. We revised the abstract and incorporated the purpose using properly structured English.

Abstract: Shading is an important technique to protect tea plantation under abiotic stresses. In this study, we analyzed the effect of shading (SD60% shade vs. SD0% no-shade) on physiological attributes and proteomic analyses of tea leaves in November and December during low-temperature. The results revealed that shading protected the tea plants, including soil plant development analysis (SPAD), photochemical efficiency (Fv/Fm) and nitrogen content (N) in November and December. The proteomics analysis of tea leaves was determined using tandem mass tags (TMT) technology and the total numbers of proteins 7263 were accumulated. Further, statistical analysis and fold change of significant proteins (FC < 0.67 and FC > 1.5 p < 0.05) revealed 14 DAPs, 11 increased and 3 decreased in November (nCK_vs_nSD60), 20 DAPs, 7 increased and 13 decreased in December (dCK_vs_dSD60), and 12 DAPs, 3 increased and 9 decreased in both November and December (nCK_vs_nSD60). These different accumulated proteins (DAPs) were dehydrins (DHNs), late-embryogenesis abundant (LEA), thaumatin-like proteins (TLPs), glutathione S-transferase (GSTs), gibberellin-regulated proteins (GAs), proline-rich proteins (PRPs), cold and drought proteins (CORA-like), and early light-induced protein 1 have been found in the cytoplasm, nucleus, chloroplast, extra cell, and plasma membrane and functioned in, catalytic, cellular, stimulus-response and metabolic pathways. In conclusion, the proliferations of key proteins were triggered by translation and posttranslational modifications, which might sustain membrane permeability in tea cellular compartments and could be responsible for tea protection under shading during low-temperature. This study aimed to investigate the impact of the conventional breeding technique "shading" and modern molecular technologies "proteomics" on tea plants, for the development of the growth and protection of new tea cultivars.

  1. The subheadings of the results must be changed such as 2.1.1. Soil Plant Analysis Development, 2.1.2 Photochemical Efficiency and Nitrogen Determination Analysis and so on….

Reply: Thank you. The section has been revised in accordance with your comments.

  1. Results

2.1. Physiological Attributes

2.1.1. Effect of Shading on Soil Plant Analysis Development in November and December during low-temperature

2.1.2. Effect of Shading on Photochemical Efficiency and Nitrogen Determination Analysis in November and December during low-temperature

2.1.3. TMT proteomic analysis of tea leaves

2.1.4. Effect of Shading on Differentially Accumulated Proteins in November and December during low-temperature

2.1.5.  PCA, GO, and KEGG Enrichment Analysis of Differentially Accumulated Proteins in November and December during low-temperature

2.1.6. Subcellular localization of Differentially Accumulated Proteins in November and December during low-temperature

2.1.7. Effect of Shading on Differentially Accumulated Proteins in November during low-temperature

2.1.8. Effect of Shading on Differentially Accumulated Proteins in December during low-temperature

2.1.9. Effect of Shading on Differentially Accumulated Proteins during both November and December

2.1.10. qRT-PCR authentication of differentially accumulated proteins during November and December

  1. In Figure 4, the title inside the figure like subcellular localization needs to modify i.e. subcellular localization protein like that. 

Reply: Changed as needed.

  1. In figure 5, mentions the genes acronym and italic form. In the caption write in detail.

Reply: We have converted all the gene acronyms to italic format and included more information in the caption.

Figure 5. Validation of differentially accumulated proteins such as; Alcohol dehydrogenase, PDZ domain-containing protein, Glutathione transferase protein, Cold and drought-regulated protein CORA-like, Early light-induced protein 1, chloroplast-like and the relative expression of their homologous genes were confirmed by qRT-PCR analysis. The significance difference is indicated by an asterisk and ns represent no-significance differences among all treatments. Data were represented as means and SD from six biological replicates (* p < 0.05).

  1. Discussion must be reduced and not more than 1500 words.

Reply: Thank you, we have minimized the discussion section and ensured that the word count does not exceed 1500.

  1. Expression analysis by qRT-PCR, explained elaborately.

Reply: Thanks, and elaborated. 

We used qRT-PCR of homologous genes of differently accumulated proteins enriched in significant metabolic pathways in the control groups and under shade during low temperatures to validate the results of the proteomics data (Figure 5). The effect of shading enhanced the expression of alcohol dehydrogenase (ADH), nevertheless, there were no significant variations identified across all groups during low-temperatures. However, the relative expression level of proteins containing PDZ domains (PDZ) exhibited substantial differences between December and November during low-temperature conditions among all groups. The levels of glutathione transferase proteins (GSTs) showed no significant differences across all groups in both November and December. However, there were notable distinctions observed between the shade and control groups in terms of the levels of cold and drought protein CORA-like (COR) during the month of November. The expression of another key early light-induced protein 1, chloroplast-like (ELIP) was reduced under shade in December compared to control groups. However, the relative expression levels between control groups and those under shade in November were satisfactory even in low-temperature conditions. The relative expression of another glutathione S-transferase (GSTs) was shown to have decreased in both the shading and control groups in November with no significant changes were noticed. However, a significant difference was observed between the shade and control groups in December under low-temperature conditions. The differences in regulations of these six randomly selected proteins and variations in their relative expression were caused by the transcriptional and translational levels under shade and no-shade control plants during low-temperature.

[We appreciate your comments and suggestions, and we hope that the manuscript now meets your expectations].

                                                                                     THANK YOU!

Round 3

Reviewer 3 Report

Comments and Suggestions for Authors

4.2. Soil Plant Analysis Development

Soil and plant analysis development (SPAD) values and nitrogen contents (N contents) of the tea leaves were determined using a plant nutrient meter (TYS-4N, Topo Yun- 839 nong Technology Co., Ltd., Zhejiang, China) according to our previous work [1,5]. 

Soil and plant analysis development values and nitrogen contents of the tea leaves  were determined using a plant nutrient meter (TYS-4N, Topo Yunnong Technology Co., Ltd., Zhejiang, China) according to our previous work [3,16].

4.3. Photochemical Efficiency Analysis 

The photochemical efficiency was measured in this study. In brief, a portable photo synthesis system (Li-6400XT, LI-COR, Inc., Lincoln, NE, USA) was used to measure chlorophyll fluorescence. The fourth developed leaf from the shoot tip was acclimated in the dark for 30 minutes under three shading groups compared with no-shade control plants during low temperatures and the photochemical efficiency of photosystem II (Fv/Fm) and  nitrogen content was calculated according to [1,5].

The fourth developed leaf from the  shoot tip was acclimated in the dark for 30 minutes under three shading groups compared 853 with no-shade control plants during low temperatures and the photochemical efficiency of photosystem II and nitrogen content was calculated according to [3,16].

Are there any differences between them? How does nitrogen content determination come from both of them?

The results subheading still not OK.

Actually, the authors are not careful and serious about their writing.

Comments on the Quality of English Language

Need serious about writing

Author Response

Comments and Suggestions for Authors

4.2. Soil Plant Analysis Development

Soil and plant analysis development (SPAD) values and nitrogen contents (N contents) of the tea leaves were determined using a plant nutrient meter (TYS-4N, Topo Yun- 839 nong Technology Co., Ltd., Zhejiang, China) according to our previous work [1,5]. 

Response: Thanks for your comments, but we don`t understand from where you copied reference [1,5]? In both sections 4.2 and 4.3. We added and improved the content of manuscript in different sections so the reference was already changed and updated in Review Report (Round 2) of revised manuscript.

In the new version of second round revised manuscript the reference should be [3,16] in the main text of heading 4.2. and 4.3. We highlighted the reference [3,16] in the reference section for your convenience in bibliography as a green color. Thank you.

Soil and plant analysis development values and nitrogen contents of the tea leaves were determined using a plant nutrient meter (TYS-4N, Topo Yunnong Technology Co., Ltd., Zhejiang, China) according to our previous work [3,16].   

4.3. Photochemical Efficiency Analysis 

The photochemical efficiency was measured in this study. In brief, a portable photo synthesis system (Li-6400XT, LI-COR, Inc., Lincoln, NE, USA) was used to measure chlorophyll fluorescence. The fourth developed leaf from the shoot tip was acclimated in the dark for 30 minutes under three shading groups compared with no-shade control plants during low temperatures and the photochemical efficiency of photosystem II (Fv/Fm) and nitrogen content was calculated according to [1,5].

The fourth developed leaf from the shoot tip was acclimated in the dark for 30 minutes under three shading groups compared 853 with no-shade control plants during low temperatures and the photochemical efficiency of photosystem II and nitrogen content was calculated according to [3,16].

Response: Once again Sorry we still did not find 853 as you mentioned above in your comment? because in Review Report (Round 2) of revised manuscript we didn`t find anything 853. We wondering you downloaded the latest version of (Round 2) manuscript file from website? Because the total lines of manuscript are 718.  Please check it again. Thank you.

Response: Thanks Yes, this is correct references [3,16]. Of second round uploaded revised manuscript version. We analysis these parameters according to our previous study already published in two well famous reputed international Journals. Such as; Frontiers in Science and Horticulturae MDPI. So, in this present manuscript we already mentioned according to [3,16] of our previous published studies. we used same methodology and experimental design etc. for these parameters according to our previous published papers Zaman et al., 2022 and Zaman et al., 2023. For your convivence and understanding we are providing you the references of these two published papers and kindly you check it with patience below.

Are there any differences between them? How does nitrogen content determination come from both of them?

Response: Thanks. Actually, Soil Plant Analysis Development and Nitrogen contents were determined by same equipment. Because the chlorophyll present in plant leaves is closely related to the nutritional condition of the plant. The measurement of SPAD values will increase in proportion to the amount of nitrogen content present in tea leaf. In fact, it depends on plant cultivar, geographical condition and other environmental factors are associated with SPAD values in plants. For a particular plant cultivar, a higher SPAD value indicates a healthier plant. Thanks.   

However, the photochemical efficiency (Fv/Fm) was determined and calculated by another instrument and methods as mentioned below with references. Thanks.

In this third round of your comments and suggestion. We again modified 4.2 and 4.3 both headings for more clearance. Kindly find the content of 4.2 and 4.3 with headings below.

4.2. Soil Plant Analysis Development and Determination of Nitrogen Content

Soil Plant Analysis Development of the tea leaves were measured in this study. In brief, the fully expanded uppermost leaves of randomly selected plants under shading and no-shade plants were determined using (SPAD 502 Meter, Minolta Corporation, Tokyo, Japan) under low-temperature [3,16] and values of Soil Plant Analysis Development (SPAD) were recorded according to [40]. The nitrogen content of tea leaves under shading and no-shade plants were also determined according to [3,16].

4.3. Photochemical Efficiency Analysis

Photochemical efficiency was also investigated in this work under shading and no-shade plants during low-temperature by sing a portable photosynthesis system (Li-6400XT, LI-COR, Inc., Lincoln, NE, USA). In short, fully expanded leaves from the shoot tip were adapted in the dark for 30 min prior to measurement. The maximum photochemical efficiency of photosystem II (Fv/Fm) was measured according to [3,16].

Given References:

[3] Zaman, S.; Shen, J.; Wang, S.; Song, D.; Wang, H.; Ding, S.; Pang, X.; Wang, M.; Sabir, I.A.; Wang, Y. Effect of Shading on Physiological Attributes and Comparative Transcriptome Analysis of Camellia Sinensis Cultivar Reveals Tolerance Mechanisms to Low Temperatures. Frontiers in Plant Science 2023, 14.

[16] Zaman, S.; Shen, J.; Wang, S.; Wang, Y.; Ding, Z.; Song, D.; Wang, H.; Ding, S.; Pang, X.; Wang, M. Effects of Shading Nets on Reactive Oxygen Species Accumulation, Photosynthetic Changes, and Associated Physiochemical Attributes in Promoting Cold-Induced Damage in Camellia Sinensis (L.) Kuntze. Horticulturae 2022, 8, 637, doi:10.3390/horticulturae8070637.

[40] Yang, H.; Li, J.; Yang, J.; Wang, H.; Zou, J.; He, J. Effects of Nitrogen Application Rate and Leaf Age on the Distribution Pattern of Leaf SPAD Readings in the Rice Canopy. PLOS ONE 2014, 9, e88421, doi: 10.1371/journal.pone.0088421.

The results subheading still not OK.

Response: Thanks. We made much modification. Once again, we changed and updated the results and subheadings numbers. Also, we changed and modified the content in results section. Thank you for understanding.

2.1. Effect of Shading on Soil Plant Analysis Development and Determination of Nitrogen Content in November and December during low-temperature

The effect of shade on soil plant analysis development of tea leaves was investigated in this study during the low-temperature. In December, during low-temperature conditions, no major differences were detected between the control group and the shading group. However, there were notable differences seen between the (nCK_vs_nSD60) conditions in November. Interestingly, it was revealed that 60% shade (Nsd60) led to a slight improvement in SPAD values compared to the control (nCK) plants in November at low-temperature (Figure 1A). The effect of shading on the nitrogen content of tea leaves were also measured in November and December during-temperature. There were no significant differences were observed between (dCK_vs_dSD60) in December during low-temperature. However, 60% of shading (Nsd60) increased the content of nitrogen compared to control (nCK) in November during low-temperature (Figure 1C).

2.2. Effect of Shading on Photochemical Efficiency in November and December during low-temperature

The effect of shading on photochemical efficiency of tea leaves was also calculated under shade during low-temperature. When compared to the control group, the photochemical efficiency was preserved by 60% shade in November. However, low-temperature affected the photochemical efficiency of un-shade control plants. The photochemical efficiency was also protected by shading (Dsd60) as compared to control (dCK) group in December during low-temperature (Figure 1B).

Also, this time for your record we used the same pattern of headings numbers (2.1., 2.2. and so on) in results section as recently published paper in the same issue by:

Zhang, Q.; Zhang, Y.; Wang, Y.; Zou, J.; Lin, S.; Chen, M.; Miao, P.; Jia, X.; Cheng, P.; Pang, X.; et al. Transcriptomic Analysis of the Effect of Pruning on Growth, Quality, and Yield of Wuyi Rock Tea. Plants 2023, 12, 3625. https://doi.org/10.3390/plants12203625. Kindly you can check it.

Actually, the authors are not careful and serious about their writing

Response: Many thanks for your encouraging remarks.

Note: We attempted to revise the paper based on 3 reviewers' comments and suggestions in round one. We modified and updated the (Abstract, Introduction, Results, Discussion, Material and Methods, Conclusion, references in text and Bibliography). Similarly, round 2 and round 3 comments and suggestions by you as a (Reviewer 3) also taken seriously and with careful consideration. Kindly download the latest version of (revised manuscript of round 3rd) with comments and suggestion.

                                                                                    Thank You!

Round 4

Reviewer 3 Report

Comments and Suggestions for Authors

Figure 6. Overview of the physiological attributes differentially accumulated proteins under shading during low-temperature.

This figure does not reflect current findings.

Comments on the Quality of English Language

OK

Author Response

Dear Editor and Reviewers;

Thank you very much for your kind help and thoughtful consideration throughout the manuscript. The comments and suggestions provided by esteemed reviewers greatly enhanced the quality of our manuscript.

Comment:

Figure 6. Overview of the physiological attributes differentially accumulated proteins under shading during low-temperature.

This figure does not reflect current findings.

Response: We have replaced the previous figure with an updated and altered version. The revised version of figure 6 clearly reflect the overall finding of present study.
